# Effect of Digital-Based Self-Learned Educational Intervention about COVID-19 Using Protection Motivation Theory on Non-Health Students’ Knowledge and Self-Protective Behaviors at Saudi Electronic University

**DOI:** 10.3390/ijerph192214626

**Published:** 2022-11-08

**Authors:** Samiha Hamdi Sayed, Mohammed Al-Mohaithef, Wafaa Taha Elgzar

**Affiliations:** 1Department of Public Health, College of Health Sciences, Saudi Electronic University, Damman 34223, Saudi Arabia; 2Department of Community Health Nursing, Faculty of Nursing, Damanhour University, Damanhour 22511, Egypt; 3Department of Maternity and Childhood Nursing, Nursing College, Najran University, Najran 66271, Saudi Arabia; 4Department of Obstetrics and Gynecology Nursing, Faculty of Nursing, Damanhour University, Damanhour 22516, Egypt

**Keywords:** COVID-19, motivation, knowledge, behavior, students

## Abstract

Background: The COVID-19 pandemic has disastrous impacts that impose the cultivation of knowledge and motivation of self-protection to foster disease containment. Aim: Evaluate the effect of digital self-learned educational intervention about COVID-19 using the protection motivation theory (PMT) on non-health students’ knowledge and self-protective behaviors at Saudi Electronic University (SEU). Methods: A quasi-experimental study was accomplished at three randomly chosen branches of SEU (Riyadh, Dammam, Jeddah) using a multistage sampling technique to conveniently select 219 students. An electronic self-administered questionnaire was used, which included three scales for assessing the students’ knowledge, self-protective behaviors, and the constructs of the PMT. The educational intervention was designed using four stages: need assessment, planning, implementation, and evaluation. A peer-reviewed digital educational content was developed after assessing the participants’ educational needs using the pretest. Then, distributed through their university emails. A weekly synchronous Zoom cloud meeting and daily key health messages were shared with them. Finally, the post-test was conducted after two months. Results: The mean participants’ age (SD) among the experimental group was 28.94 (6.719), and the control group was 27.80 (7.256), with a high female percentage (63.4%, 73.8%) and a previous history of direct contact with verified COVID-19 patients (78.6%, 69.2%), respectively. A significant positive mean change (*p* = 0.000) was detected in the total COVID-19 knowledge of the experimental group post-intervention, either when it was adjusted for the covariates effect of the control group (F_1_ = 630.547) or the pretest (F_1_ = 8.585) with a large effect size (η^2^ = 0.745, η^2^ = 0.268, respectively). The same was proved by the ANCOVA test for the total self-protective behaviors either when it adjusted for the covariates effect of the control group (F_1_ = 66.671, *p* = 0.000) or the pretest (F_1_ = 5.873, *p* = 0.020) with a large effect size (η^2^ = 0.236, η^2^ = 0.164, respectively). The ANCOVA test proved that post-intervention, all the PMT constructs (perceived threats, reward appraisal, efficacy appraisal, response cost, and protection intention) and the total PMT score were significantly improved (*p* = 0.000) among the experimental group either when adjusted for the covariates effect of the control group (F_1_ = 83.835) or the pretest (F_1_ = 11.658) with a large effect size (η^2^ = 0.280, η^2^ = 0.561, respectively). Conclusions: The digital PMT-based self-learned educational intervention effectively boosts non-health university students’ COVID-19 knowledge, protection motivation, and self-protective behaviors. Thus, PMT is highly praised as a basis for COVID-19-related educational intervention and, on similar occasions, future outbreaks.

## 1. Introduction

The world is fighting coronaviruses and their related respiratory illness. In 2003, there was the severe acute respiratory syndrome (SARS-CoV) and the 2012 Middle East respiratory syndrome (MERS-CoV). Then, in December 2019, an extraordinary zoonotic coronavirus named SARS-CoV2 first emerged as a respiratory illness outbreak in Wuhan city/China [1,2]. It is formally known as Coronavirus Disease-2019 (COVID-19). It was announced by the World Health Organization (WHO) on 30 January 2020 as a public health emergency and on 11 March 2020 as a global pandemic. It is highly contagious in nature, either by direct respiratory droplets or indirect environmental contact [3,4]. In 2022, the WHO global estimate confirmed that COVID-19 cases are approaching 590 million, with more than 6 million deaths. It registered a surge in the incidence of weekly cases in European, the Americas, the Western Pacific, South-East Asia, the Eastern Mediterranean, and the Africa Regions, respectively [5]. Its manifestations can go undetected or range from mild to moderate, with the possibility of home recovery (80%) [6]. The disease can result in severe complications with higher mortality, especially among high-risk individuals such as the elderly, pregnant women, diabetics, cardiovascular and cancer patients, immunocompromised, smokers, substance abusers, obese persons, and those with pre-existing chronic respiratory diseases. It might have serious multi-organ effects that exist four to twelve weeks after recovery or several months, called post-COVID syndrome or long-haul [6,7,8].

Since no definitive treatments are available for COVID-19, fostering proactive engagement in preventive practices is the core of establishing successful disease containment strategies. This can be achieved through proper education and motivation to empower the population to comply with these recommended behaviors. Social distancing (at least 1 m), respiratory etiquette, frequent hand washing or sanitizing, face masking, surfaces disinfection with alcohol-based disinfectants, and averting poorly ventilated and overcrowded areas were the evidenced coronavirus preventive behaviors by The WHO and Centres for Disease Control and Prevention (CDC) [4,9]. Distinctly, perceiving the associated health threats with this novel virus along with the motivation for self-protection, can play a central role in boosting compliance with such self-protection behaviors; this issue was highlighted by the PMT [10]. Ronald Rogers first generated the Protection Motivation Theory (PMT) in 1975 and revised it in 1983 to explain the cognitive process of behavioral change through two main pathways; threat appraisal (disease) and coping appraisal (recommended actions), which can lead to adaptive or maladaptive behavioral responses. In this regard, a fear appraisal from the identified health threat plays an essential role in predicting and motivating protective behaviors by influencing the individuals’ attitudes, emotions, and perception of the threat’s severity. This further best predicts engagement in protective health behaviors [11,12]. 

The PMT elaborated that the individuals’ readiness and intention to initiate a health-promoting behavior or avoid a health-risk behavior is mainly determined based on four factors arranged over the two pathways. The threat appraisal pathway comprises two factors: an individual’s estimation of the disease’s severity (perceived severity) and the likelihood of contracting this disease (perceived vulnerability). The threat perception can be affected by many environmental and personal factors that can influence the level of engagement in the proposed health behaviors (extrinsic and intrinsic reward). The coping appraisal pathway includes another two factors; outcome worthiness or the individual’s anticipation that engaging in the suggested behaviors can eliminate the threat (response efficacy). The individual’s belief in his capabilities to accomplish these suggested behaviors (self-efficacy). Conversely, suppose the costs associated with the recommended health behaviors concerning the surrounding physical, emotional, and social context (response cost) outweigh its benefits (response efficacy), the individuals will not be motivated to engage in these behaviors or maintain them for a long time [10,12,13]. In the context of COVID-19, PMT can assist in exploring that those who are expected to engage in the suggested preventing behaviors will be those who perceive that they have high vulnerability to coronavirus and believe that it is a life-threatening infection. The pandemic is also associated with great public stress and anxiety, which amplifies the fear level and increase the probability of adoption of preventive behaviors. Furthermore, the belief that engagement in these proposed preventive behaviors and their ability to do so will substantially lessen the infection risk. Moreover, the perception of having supportive environmental circumstances and policies can also facilitate compliance with such behaviors [13,14,15]. 

Moreover, the recurrent waves of coronavirus infections created a great surge of rumors and misleading information about this novel virus called “infodemic”. Thus, it becomes of utmost importance to develop evidence-based education [16,17]. A theory-based intervention using information and communication technology (ICT) and self-education proves robust evidence in the latest globalization era. Self-learning and digital-based education, as an example of ICT, offer several benefits. They have more flexibility, accessibility, and low cost, which can help in cultivating knowledge and behavioral change as guided by self-motivation, not coercive attendance of predesigned sessions [18,19,20]. In this study, the investigators apply this approach guided by the PMT using four stages approach: need assessment, planning, implementation, and evaluation. A peer-reviewed educational content (PowerPoint presentation and a recorded session using a blackboard) was developed after assessing the participants’ educational needs. Then, they were made available to the participants through their official university emails. A synchronized availability of the investigators through Zoom cloud meetings was undertaken weekly throughout the study duration to answer their questions and concerns. A key health message about COVID-19 was shared with them daily to maintain contact, and engagement, boost their self-efficacy and intention for self-protection and turn it into action.

### 1.1. Significance of the Study

Given the novelty of coronavirus, most of the previous evidence was descriptive, with general deficient knowledge about COVID-19 in many countries [6,18]. In addition, evidence showed little concern for educational trials to examine the effect of educational intervention on the pandemic response. Two recent educational interventions proved the effectiveness of educational intervention during the COVID-19 pandemic and recommended its incorporation as a main element of the potential outbreak responses in the future [19,20]. Thus, the current study intends to examine the effect of digital-based self-learned educational intervention about COVID-19 using the PMT on non-health students’ knowledge and self-protective behaviors at Saudi Electronic University.

### 1.2. Hypotheses

The non-health colleges students who received the digital self-learned educational intervention about COVID-19 using the PMT exhibited higher scores than the control group for all the study variables: The COVID-19 knowledge;The self-protective behaviors;The PMT components (perceived vulnerability, perceived severity, fear, intrinsic reward, extrinsic reward, response efficacy, self-efficacy, response cost, and protection intention).

## 2. Materials and Methods

### 2.1. Study Design and Setting

A quasi-experimental study registered in the WHO Registry Network through the Iranian Registry of Clinical Trial [IRCT20210131050192N3]. It was conducted at three randomly selected branches of Saudi Electronic University (SEU), namely, Riyadh, Dammam, and Jeddah, between May and December 2021.

### 2.2. Participants

A multistage sampling technique was utilized. First, three branches from SEU were randomly chosen (Riyadh, Dammam, and Jeddah). Second, in each branch, the three available non-health colleges were included to avoid contamination of the sample (administrative and financial sciences, sciences and theoretical studies, and computing and informatics colleges). Third, a convenient sample from the students enrolled in each selected college was incorporated. The inclusion criteria included enrolled students in SEU with a willingness to participate in the study. However, students enrolled in health colleges or who previously participated in any COVID-19 educational program were excluded. The sample size was estimated based on the following parameters: standardized effect size (0.5), a standard deviation of the outcome (1.0), the proportion of subjects in case and control groups (0.5), β/type II error (0.2), α (two-tailed)/type I error = 0.05 [21,22]. This yielded a minimum sample size per group of 63 students. After considering the design effect to adjust for the cluster size, the final required sample size per group is 104 giving a total sample size for both groups of 208 students. The final overall sample size was 219 distributed over the experimental (112) and control (107) groups. The participants were incorporated into the study using the next follow chart (Figure 1):

### 2.3. Survey Development

The investigators generated an electronic self-administrated questionnaire after reviewing the related recent literature. It involved four main parts: 

**Part I: Basic Data:** age, sex, academic level, marital status, occupation, residence, previous work in health institutions, and diagnosis of a family member with COVID-19. 

**Part II: The COVID-19 knowledge scale:** comprised 27 multiple-choice questions related to COVID-19 manifestations, transmission, risk factors, prevention, vaccination, complications, emergency signs, and management [19,20]. The participants were scored “one” for the “yes” answer and “zero” for the “no” or “do not know” answer. The total knowledge score ranged from 0 to 27, where higher scores reflected better COVID-19-related knowledge. A question about the COVID-19 sources of information was added but not included in the total knowledge score.

**Part III: Self-Protective Behaviors Scale:** it was developed based on the recommended preventive behaviors of coronavirus infection by the WHO and CDC [3,8]. It contained six items rated on a three-point Likert scale: “always” (3), “occasionally” (2), and “never” (1). The total score ranged from 6 to 18, where higher scores signify better COVID-19 self-protective behaviors.

**Part IV: PMT Constructs Scale:** it was developed by the researchers based on PMT-related evidence [10,12,23,24] to assess the protection motivation factors regarding the COVID-19 pandemic. It comprised 28 items distributed as follows: perceived threats (vulnerability, severity, and fear), reward appraisal (intrinsic and extrinsic), efficacy appraisal (response efficacy and self-efficacy), and response cost, with three items per each, while the protection intention had four items. In each item, the participants chose one of five alternatives from “strongly agree” (5) to “strongly disagree” (1). For response cost items, the scoring was reversed. The total scores were calculated for each construct and the total PMT, where a higher score indicated higher self-protection motivation. The English version of the used questionnaire is available as a Appendix A. 

### 2.4. Instrument’s Validity and Reliability

The instrument was designed by the researchers based on relevant, credible evidence and translated into Arabic with back translation by a different researcher to guarantee its accuracy. Its content was agreed upon by a panel of six experts in the field who also rated the items based on their wording, ordering, construct relevancy, and scoring. Based on their feedback, the questionnaire was revised and modified. It showed an excellent overall Scale–Content Validity Index (S–CVI = 0.87) and per item (I–CVI) that ranged between 0.7 to 1.0. The instrument‘s discriminative ability was evaluated using Pearson’s correlation coefficient between items and the respective scales. Thus, two items were removed due to low coefficient (<30) and lack of significance (*p* > 0.05). The instrument’s construct validity was confirmed by the confirmatory factor analysis, which showed a good fit using the comparative fit index (CFI > 0.90). The instrument reliability was assessed using Cronbach’s Alpha (α) coefficient test (α for part II = 0.72, part III = 0.84, and part IV = 0.81). 

### 2.5. Pilot Study

The instrument was piloted on 10% of the participants who were omitted from the main study sample. It aimed to ascertain its clarity, applicability, and validity, wherefore the crucial modifications were conducted.

### 2.6. Fieldwork

The data was collected from the start of September until the end of December 2021. The researchers developed a survey link using the Survey Monkey program and sent it to the SEU students. After taking students’ consent for the study participation, an assessment question of the pre-determined exclusion criteria was set. Consequently, the students were assigned either to the experimental or the control group based on the pre-prepared list. 

The educational intervention was designed and carried out for the experimental group based on the PMT through four consecutive phases: 

**Needs assessment:** it aimed to assess the students’ background knowledge, current self-protection behaviors, and protection motivation factors during the COVID-19 pandemic using the developed instrument as a pretest. The results were analyzed to reveal the students’ educational needs and facilitate the post-test comparison to be used as a base for developing the program educational content and the supportive media.

**Planning:** based on the results of the need assessment phase, considering the relevant literature, the investigators designed a digital educational intervention based on PMT. It included a PowerPoint presentation and an education session about COVID-19 was recorded using the blackboard as the official learning management system in SEU. It contained basic information about COVID-19, such as modes of transmission, signs, symptoms, vulnerable groups, danger signs, and self-protection behaviors. Then, the developed content was independently evaluated by three external reviewers and the required modifications were made. This intervention aimed to enhance COVID-19 knowledge and foster a positive attitude and beliefs about its preventive behaviors. It also helped boost students’ self-efficacy and self-protection intention by deploying specific measures to turn this behavioral intention into action to overcome the potential barriers to the recommended behaviors. The researchers collected the students’ phone numbers to develop a WhatsApp group for the experimental group to facilitate communication. 

**Implementation:** After gaining the ethical and data collection approvals from SEU and approval of the program’s content and the digital media by the reviewers, the PowerPoint presentation, and the recorded educational session were sent to the students through their official university emails. A key health message about COVID-19 was sent to them through WhatsApp daily to maintain contact, and engagement, boost their self-efficacy and intention for self-protection and turn it into action. A weekly synchronous availability of the investigators through Zoom cloud meetings was done after confirming the suitability of the timing with the group through WhatsApp, to answer their questions, discuss concerns, and correct any misinformation.

**Evaluation:** the follow-up of the experimental group was conducted after two months using the same pretest tools (Part II, III, IV) to assess the students’ COVID-19-related knowledge, self-protection behaviors, and PMT constructs. 

**Regarding the control group:** an online pretest was done and repeated two months later, using the same pretest tools used for the experimental group. Ethically, the researchers shared the PowerPoint presentation and the recorded session with the control group after the study completion. 

### 2.7. Statistical Analysis

After the end of data collection, it was entered into the Statistical Package of Social Science (SPSS) software, version 26. Descriptive statistics such as numbers, percentages, arithmetic mean, and standard deviation were employed to describe and summarize data. The significance of categorical variable differences between groups was investigated using the Chi-square test or Fisher’s exact test. Differences in students’ knowledge, protective behaviors, and PMT constructs before and after the educational intervention and between experimental and control groups were examined by the analysis of covariance (ANCOVA) test to examine the effect size of the intervention and control the effects of the covariates in the context of losing randomization. The basic rules of thumb followed for measuring the effect size using the partial eta squared (η^2^) were small (0.01), medium (0.06), and large (0.14) effects [25]. The cut-off value of the significance level was set as *p* ≤ 0.05 

### 2.8. Ethical Considerations

The study was approved by the ethical committee of SEU (IRB number: SEUREC-CHS21124). Formal approval for study conduction was obtained from each college dean after explaining its aim. Electronic informed consent was taken from each student. All data were kept confidential and used only for research purposes. Moreover, the students were informed about their right to unconditional withdrawal from the study at any time.

## 3. Results

### 3.1. Participants’ Basic Data and Sources of Information about COVID-19

Table 1 reveals no statistically significant differences between the experimental and control groups related to basic data, indicating homogeneity. The mean participants’ age was 28.94 years among the experimental compared to 27.80 in the control group. Females signified a higher percentage in the experimental (63.4%) and control (73.8%) groups. Students at the higher academic level (5–8) stood for 92.0% and 81.0% of the experimental and control group, respectively, and 49.1% of the experimental and 59.8% of the control group did not work in health institutions. As expected, 78.6% of the experimental group and 69.2% of the control group have a history of direct contact with verified COVID-19 patients. 

The most frequent sources of information about COVID-19 pre-intervention, among both the experimental and control groups, were the Saudi ministry of health (87.5%, 90.7%), healthcare staff (73.2%, 75.7%), and international official health websites (68.8%, 71.0%), respectively. Around one-third (31.3%, 36.4%) of the experimental and control, groups use social media, while (25.9% and 29.9%) of them depend on friends and relatives, respectively.

### 3.2. The PMT Constructs Scores Pre- and Post-Intervention for the Two Groups

Table 2 portrayed a statistically significant mean change in all the PMT constructs post-intervention among the experimental group; perceived threats (vulnerability, severity, and fear), reward appraisal (intrinsic and extrinsic), efficacy appraisal (response efficacy, and self-efficacy), response cost, and protection intention (*p* < 0.05). This was proven by the ANCOVA test either when the total PMT constructs score was adjusted for the covariates effect of the control group (F_1_ = 83.835, *p* < 0.05) or the pretest (F_1_ = 11.658, *p* < 0.05). A large effect size (η^2^ = 0.561) among the experimental group and a large variance (η^2^ = 0.280) between the two groups were attributed to the educational intervention. 

### 3.3. The COVID-19 Knowledge Scores Pre- and Post-Intervention for the Two Groups

Table 3 portrays a statistically significant mean change in the total COVID-19 knowledge of the experimental group post-intervention (18.63 ± 1.433 versus 24.33 ± 1.747). This was proven by the ANCOVA test either when the total knowledge score was adjusted for the covariates effect of the control group (F_1_ = 630.547, *p* < 0.05) or the pretest (F_1_ = 8.585, *p* < 0.05). A large effect size (η^2^ = 0.268) among the experimental group and a large variance (η^2^ = 0.745) between the two groups was attributed to the educational intervention. 

### 3.4. Self-Protective Behaviors Scores Pre- and Post-Intervention for the Two Groups

Table 4 portrayed a statistically significant mean change in the total self-protective behaviors of the experimental group post-intervention (14.750 ± 0.729 versus 17.12 ± 2.636). This was proven by the ANCOVA test either when the total self-protective behaviors score was adjusted for the covariates effect of the control group (F_1_ = 66.671, *p* < 0.05) or the pretest (F_1_ = 5.873, *p* < 0.05). A large effect size (η^2^ = 0. 164) among the experimental group and a large variance (η^2^ = 0. 236) between the two groups were attributed to the educational intervention. 

## 4. Discussion

The current study verified a significant improvement in the COVID-19 knowledge post-intervention among the experimental group with a large effect size (η^2^ > 0.14). These findings reflected the proper designation, implementation, and efficacy of the digital-based self-learned educational intervention using PMT constructs. Similarly, an Egyptian educational trial proved improvement in the participants’ knowledge and succeeded in modifying their COVID-19-related misconceptions along with a high satisfaction level with the digital intervention [19]. This parallels the KSA 2030 vision that paved the path for digital transformation, promoted and tested by the COVID-19 pandemic [26]. A Saudi study [27] also indicated a significant positive mean change in the university students’ knowledge score, which was positively correlated with their perceived severity and vulnerability, self-efficacy, perceived benefits, and barriers to COVID-19 preventive behaviors.

The present study highlighted that most of the participants rely on the Saudi ministry of health, health care staff, and international official health websites as the main sources of information about COVID-19. However, social media, friends, and relatives were still relied on by many of them. This reflected better health awareness among the Saudi population through relying on rigorous and scientific health resources, which can have a crucial role in shaping their attitudes and health behaviors. Thus, informal sources (i.e., social media) should be targeted and prioritized by the government to ensure its credibility and avoid the “infodemic” associated with the pandemic [28]. Conveniently, a novel Chinese study proved that obtaining COVID-19-related information from health personnel was correlated with higher self-efficacy, response efficacy, and knowledge about COVID-19 vaccination; however, obtaining information from colleagues was associated with lesser efficacy and knowledge [17]. A European study also revealed a lower level of trust for officials, mass media, and social media than the medical scientists and professionals who were considered the most credible. It also confirmed that the perceived credibility of COVID-19 information was linked with lower negative emotional responses and higher adherence to self-protective behaviors [29]. 

The present study proved the effectiveness of the PMT in improving the participants’ self-protective behaviors (η^2^ > 0.14) with significant improvements in all the PMT constructs post-intervention: perceived threats (vulnerability and severity), reward appraisal (intrinsic and extrinsic), efficacy appraisal (response and self-efficacy), response cost, and behavior intention with a large effect size (η^2^ > 0.14). This reflects the PMT’s effectiveness in behavior modification through using motivational online interviews and intention implementation measures to increase the students’ self-efficacy to overcome barriers associated with adherence to preventive behaviors. Evidently, a recent study revealed that the PMT components predicted a high level of variance in the frequency of the protective health behaviors during waves 1 and 2 of COVID-19 [10]. In addition, significant positive correlations were proved between the COVID-19 preventive behaviors and the PMT components; perceived vulnerability and severity, response efficacy, self-efficacy, and protection motivation [24]. Moreover, a recent Indian study proved that self-efficacy and COVID-19 threat-appraisal were crucial determinants of its preventive behaviors [30]. However, up until now, the PMT has not been used as a theoretical background for educational intervention about COVID-19, our study is the pioneer in this regard.

Evidence is conclusive about the effectiveness of PMT-based educational intervention in behavior modification such as physical activity and diabetes management [31], cervical [32], and skin [33,34] cancer preventive behaviors. Moreover, three educational interventions using the PMT proved a significant positive mean change in the knowledge and preventive behaviors post-intervention for weight control [35], UV radiation exposure disorders [36], and HIV/AIDs [37]. Thus, raising awareness about the individual’s vulnerability help in boosting the threat appraisal and response efficacy to reinforce the coping appraisal for self-protection. Variations in the PMT constructs improvements were explored by a recent study which concluded that the PMT-based education is effective in improving type A influenza preventive behaviors among Iranian male students, with a significant increase in the mean scores of the perceived severity, self-efficacy, reward, and protection motivation [38]. However, it found no significant difference in the mean score of perceived vulnerability, response efficacy, response cost, and fear. This partial disagreement may be due to the different nature of the infectious disease, where, in the current study, we talk about the unprecedented COVID-19 pandemic, which can generate more fear and vulnerability and impose more restrictive protective behaviors. Besides, in this contradictory study, the subjects were male-only, where recent studies evidenced gender differences during COVID-19 pandemics in perceiving risk and complying with protective behaviors in the favor of females [39]. 

Conversely, the current study is not in line with a recent study [17] which revealed that the PMT’s components of coping appraisal (response efficacy, costs of COVID-19 vaccine, and self-efficacy for vaccination) and COVID-19 vaccination-related knowledge were not significant predictors of the motivation for COVID-19 vaccination. An Iranian study [40] also reported no significant differences between the intervention and control groups in the perceived severity, fear, response efficacy, and response cost following PMT-based education. These dissimilarities may be attributed to the difference in the target of the study as this contradictory study focus on the Pap test practice among women besides, the varied social context, study designation, and implementation. Moreover, recent evidence during the COVID-19 pandemic suggested that despite the fundamental role of encouraging the COVID-19 precautionary measures, the communication strategies should not only overemphasize their safety benefits because this may result in unrealistic optimism of their susceptibility to COVID-19, as an unintended psychological consequence. This is mainly due to being more concerned with their own behaviors regardless of the others’ actions (egocentrism) and thus feeling safer in comparison with others which may later decrease their willingness to comply with the COVID-19 preventive recommendations [41,42], hence, the educational intervention should be properly designed with an evidence-based theoretical framework such as the PMT to cut down the negative association between unrealistic optimism and self-protective behavior. 

### Strengths, Limitations, and Future Implications

The present study is a pioneer in Saudi Arabia in applying PMT to an emerging disease like COVID-19. It also was conducted in three different cities to capture a holistic image and be more representative. It aids in improving the COVID-19-related knowledge and self-protection, which may help policymakers in disease containment. However, one of the limitations of this study is the digital survey which is suitable only for literate individuals with Internet access. The convenient sample approach is a must in the current study because of the geographically scattered study areas. This issue was compensated for by real-time interactive sessions. Thus, the current study recommended boosting the digital educational intervention strategies, developing campaigns to foster awareness about the protection motivation factors using the PMT on a wider basis, and further replication of similar studies on different populations as there is scarce evidence in this regard.

## 5. Conclusions

Based on the current study findings, the null hypothesis is rejected, and the PMT was proven to be effective in improving university students’ COVID-19 knowledge, self-protective behaviors, and protection motivation. The COVID-19 knowledge and self-protective behaviors were significantly improved among the experimental group after applying the digital-based educational intervention using the PMT. All the protection motivation components were significantly improved post-intervention; perceived threats (vulnerability, severity, and fear), reward appraisal (intrinsic and extrinsic), efficacy appraisal (response efficacy and self-efficacy), response cost, and protection intention.

## Figures and Tables

**Figure 1 ijerph-19-14626-f001:**
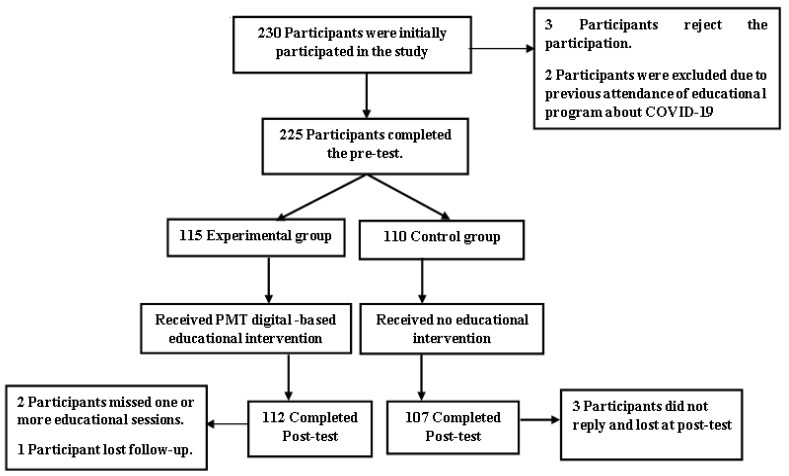
Participants’ flowchart.

**Table 1 ijerph-19-14626-t001:** Participants’ basic data and sources of information about COVID-19.

	Experimental Group N (112)	Control Group N (107)	X^2^/FET/*t*	df	*p*
	N (%)	N (%)			
**Sex**					
Male	41 (36.6)	28 (26.2)	X^2^ = 2.763	1	0.96
Female	71 (63.4)	79 (73.8)			
**Marital status**					
Married	59 (52.7)	47 (43.9)	FET = 2.999	3	0.392
Divorced	5 (4.4)	6 (5.6)			
Single	48 (42.9)	54 (50.5)			
**Academic level**					
1–4	20 (17.85)	26 (24.29)	X^2^ = 8.568	1	0.130
5–8	92 (82.15)	81 (75.21)			
**Occupation**					
Governmental	47 (42)	37 (34.6)	FET = 5.222	3	0.156
Private	9 (8)	4 (3.7)			
Free business	3 (2.7)	1 (0.9)			
Not working	53 (47.3)	65 (60.7)			
**Residence**					
Riyadh	44 (39.3)	37 (34.6)	FET = 2.045	3	0.359
Dammam	43 (38.4)	37 (34.6)			
Jeddah	25 (22.3)	33 (30.8)			
**Previous work in health institutions**					
Yes	57 (50.9)	43 (40.2)	X^2^ = 2.528	1	0.112
No	55 (49.1)	64 (59.8)			
**Direct contact with confirmed COVID-19 patient**					
Yes	88 (78.6)	74 (69.2)	X^2^ = 2.518	1	0.113
No	24 (21.4)	33 (30.8)			
**Age in years-Mean (SD)**	28.94 (6.719)	27.80 (7.256)	*t* = 1.2	217	0.231
**Sources of information about COVID-19 pre-intervention #**					
- Saudi Ministry of Health	98 (87.5)	97 (90.7)			
- Health Care Staff	82 (73.2)	81 (75.7)	X^2^ = 6.564	1	0.195
- International Health Websites	77 (68.8)	76 (71.0)			
- Official Saudi Health websites	66 (58.9)	60 (56.1)			
- Peer-Reviewed Health Journals	43 (38.4)	35 (32.7)			
- Mass media	36 (32.1)	38 (35.5)			
- Social media	35 (31.3)	39 (36.4)			
- Friends and Relatives	29 (25.9)	32 (29.9)			

X^2^: Chi-square test, FET: Fisher exact Test, *t*: independent sample *t*-test, # Responses are not mutually exclusive.

**Table 2 ijerph-19-14626-t002:** Analysis of covariance in the Mean scores of the PMT constructs pre and post PMT-based educational intervention.

PMT Constructs	Pre		Post		Reference (Control Group)			Reference (Pretest)		
	Experimental Mean (SD)	Control Mean (SD)	Experimental Mean (SD)	Control Mean (SD)	F	*p* Value	Partial η^2^	F	*p* Value	Partial η^2^
- Perceived vulnerability	8.021 (2.493)	7.22 (2.493)	10.79 (1.807)	8.98 (1.780)	F_1_ = 54.837	<0.001 *	0.202	F_1_ = 2.689	0.675	0.013
- Perceived severity	11.78 (2.074)	11.29 (2.124)	13.03 (1.896)	11.69 (1.718)	F_1_ = 27.449	<0.001 *	0.113	F_9_ = 5.109	0.030 *	0.221
- Fear	10.035 (3.048)	9.336 (3.209)	11.61 (2.170)	10.29 (2.391)	F_1_ = 21.817	<0.001 *	0.092	F_2_ = 9.442	0.002 *	0.042
**Total perceived threats**	29.812 (5.107)	27.850 (6.104)	35.428 (4.531)	30.962 (4.076)	F_1_ = 58.998	<0.001 *	0.215	F_2_ = 6.232	0.012 *	0.442
- Intrinsic reward	12.866 (1.398)	13.1 (1.687)	13.68 (1.520)	12.50 (1.403)	F_1_ = 34.731	<0.001 *	0.139	F_2_ = 12.688	0.039 *	0.194
- Extrinsic reward	12.258 (1.054)	12.196 (1.598)	13.40 (1.624)	12.00 (1.732)	F_1_ = 38.012	<0.001 *	0.150	F_9_ = 3.629	0.019 *	0.262
**Total reward appraisal**	25.125 (2.160)	24.234 (3.048)	28.080 (2.809)	24.504 (2.199)	F_1_ = 56.405	<0.001 *	0.207	F_14_ = 2.850	0.016 *	0.365
- Response efficacy	11.705 (1.305)	11.320 (1.647)	12.99 (1.872)	11.52 (1.501)	F_1_ = 40.631	<0.001 *	0.158	F_1_ = 2.180	0.141	0.010
- Self-efficacy	11.883 (1.353)	12.196 (1.538)	12.99 (1.788)	11.78 (1.803)	F_1_ = 25.841	<0.001 *	0.107	F_1_ = 2.137	0.045 *	0.025
**Total efficacy appraisal**	23.589 (2.224)	23.953 (2.991)	26.982 (3.229)	24.102 (2.381)	F_1_ = 47.952	<0.001 *	0.182	F_9_ = 3.811	0.016 *	0.030
**Response cost**	7.848 (3.252)	6.981 (3.135)	8.53 (4.025)	7.04 (3.555)	F_1_ = 8.102	<0.001 *	0.036	F_1_ = 3.658	0.007 *	0.312
**Protection intention**	14.750 (0.729)	13.59 (0.988)	17.12 (2.636)	14.47 (2.134)	F_1_ = 66.671	<0.001 *	0.236	F_1_ = 5.873	0.020 *	0.164
**Total PMT score**	101.01 (9.255)	99.25 (8.037)	114.13 (13.373)	100.27 (8.282)	F_1_ = 83.835	<0.001 *	0.280	F_1_ = 11.658	0.000 *	0.561

Note: * significant at *p* ≤ 0.05, analysis of covariance (ANCOVA) η^2^ = Eta squared (effect size).

**Table 3 ijerph-19-14626-t003:** Analysis of covariance in the mean score of COVID-19 knowledge pre and post PMT-based educational intervention.

COVID-19 Knowledge	Pre		Post		Reference (Control Group)			Reference (Pretest)		
	Experimental Mean (SD)	Control Mean (SD)	Experimental Mean (SD)	Control Mean (SD)	F	*p* Value	Partial η^2^	F	*p* Value	Partial η^2^
Mode of transmission	2.79 (0.473)	2.81 (0.0.552)	3.39 (0.543)	2.91 (0.539)	F_1_ = 106.880	<0.001 *	0.331	F_1_ = 77.248	0.000 *	0.293
Signs and symptoms	4.81 (0.456)	4.65 (0.472)	6.46 (0.747)	4.88 (0.381)	F_1_ = 365.328	<0.001 *	0.647	F_2_ = 5.472	0.020 *	0.025
High-risk groups	3.95 (0.837)	4.07 (0.756)	4.97 (0.788)	4.16 (0.881)	F_1_ = 51.250	<0.001 *	0.191	F_2_ = 4.865	0.029 *	0.032
Preventive measures	4.25 (0.511)	4.28 (0.491)	5.55 (0.627)	4.57 (0.585)	F_1_ = 145.978	<0.001 *	0.403	F_2_ = 2.744	0.099	0.013
Emergency signs	2.84 (0.393)	2.94 (0.231)	3.95 (0.263)	2.67 (0.510)	F_1_ = 527.223	<0.001 *	0.709	F_1_ = 10.81	0.002 *	0.060
Vaccination	2.78 (0.447)	0.2.93 (0.521)	4.34 (0.652)	3.10 (0.614)	F_1_ = 473.642	<0.001 *	0.687	F_1_ = 3.368	0.037 *	0.041
**Total knowledge score**	18.63 (1.433)	18.93 (1.226)	24.33 (1.747)	18.92 (1.388)	F_1_ = 630.547	<0.001 *	0.745	F_1_ = 8.585	0.000 *	0.268

* significant at *p* ≤ 0.05, analysis of covariance (ANCOVA) η^2^ = Eta squared (effect size).

**Table 4 ijerph-19-14626-t004:** Analysis of covariance in the mean scores of COVID-19 self-protective behaviors pre and post PMT-based educational intervention.

Self-Protective Behaviors	Pre		Post		Reference (Control Group)			Reference (Pretest)		
	Experimental Mean (SD)	Control Mean (SD)	Experimental Mean (SD)	Control Mean (SD)	F	*p* Value	Partial η^2^	F	*p* Value	Partial η^2^
- Wearing facemask	2.01 (0.094)	1.989 (0.168)	2.52 (0.502)	2.03 (0.166)	F_1_ = 37.504	<0.001 *	0.130	F_1_ = 0.528	0.468	0.002
- Social distancing	2 (0.232)	1.99 (0.217)	2.38 (0.486)	2.04 (0.191)	F_1_ = 45.479	<0.001 *	0.174	F_9_ = 4.223	0.020 *	0.135
- Hand washing or alcohol rub.	2.035 (0.186)	2 (0.130)	2.54 (0.500)	2.04 (0.121)	F_1_ = 124.568	<0.001 *	0.366	F_2_ = 6.112	0.004 *	0.072
- Disinfect surfaces	2 (0.355)	2 (0.275)	2.52 (0.502)	1.95 (0.212)	F_1_ = 118.304	<0.001 *	0.354	F_2_ = 3.173	0.007 *	0.052
- Respiratory etiquette	2 (0.002)	2 (10.001)	2.57 (0.497)	2.03 (0.097)	F_1_ = 43.106	<0.001 *	0.180	F_2_ = 5.688	0.042 *	0.014
- Healthy lifestyle	1.95 (0.351)	1.79 (0.347)	2.54 (0.510)	1.90 (0.362)	F_1_ = 120.006	<0.001 *	0.357	F_9_ = 4.629	0.039 *	0.027
- Hospital visiting for emergency signs	2.13 (0.332)	1.96 (0.317)	2.47 (0.376)	2.08 (0.132)	F_1_ = 65.705	<0.001 *	0.227	F_14_ = 2.944	0.040 *	0.024
- Vaccination	1.35 (0.551)	1.46 (0.447)	2.13 (0.432)	1.731 (0.372)	F_1_ = 37.742	<0.001 *	0.231	F_1_ = 0.337	0.562	0.002
**Total self-protective behaviors**	14.750 (0.729)	13.59 (0.988)	17.12 (2.636)	14.47 (2.134)	F_1_ = 66.671	<0.001 *	0.236	F_1_ = 5.873	0.020 *	0.164

Note: * significant at *p* ≤ 0.05, analysis of covariance (ANCOVA) η^2^ = Eta squared (effect size).

## Data Availability

The dataset can be accessed upon request.

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
