# Peer review of "Effect of Digital-Based Self-Learned Educational Intervention about COVID-19 Using Protection Motivation Theory on Non-Health Students’ Knowledge and Self-Protective Behaviors at Saudi Electronic University"

_ijerph, 2022, doi:10.3390/ijerph192214626_

Round 1
Reviewer 1 Report
Said and colleagues performed a quasi-experimental study to assess the effect of a digital-based self-learned education intervention about COVID-19 using protection theory on non-health students’ knowledge and self-protective behaviors at SEU. This study was generally well-conducted with a robust design. The major concern that the reviewer has while reading this manuscript is how the intervention was introduced and described, which seems supposed to be the main focus of this work but is not well-described and enough highlighted in the paper. The reviewer had the following comments regarding this concern and other minor comments/suggestions for the authors’ reference to further refine this manuscript.
1. Introduction - The authors provided thorough background information on the COVID-19 pandemic and the theoretical premise of how proper education and self-protective behaviors could foster disease containment guided by the Protective Motivation Theory (PMT). However, the necessary background information about the development of the intervention per se (i.e., the digital-based self-learned education intervention) is lacking in the Introduction. Please elaborate on the intervention program itself including design, necessary details about the program itself, how it was administered in the institution, etc.
2. Abstract - Related to #1. There was a lack of description of the administration of digital self-learned education intervention about COVID-19 in the Methods section of the abstract.
3. Material and Methods - Consider providing the list of key questions included in the questionnaire as supplemental material (e.g. Part II-IV). It would be also helpful to specify how individual questions were mapped to the pre-specified survey constructs.
4. Material and Methods - The reviewer was wondering whether the educational intervention (2.5.) should be moved before the survey development (2.3.).
5. Material and Methods - currently the statistical analyses were described under 2.5. Pilot Study and Data Collection. The reviewer would recommend having a standalone section for data analysis to avoid confusion.
6. Were any sociodemographic characteristics analyzed in association with the outcomes of interest related to knowledge and self-protective behaviors?
Minor:
7. Page 2, para 2: spell out PMT when it first appeared in the main text.
8. Please confirm all P values of 0.000. The reviewer recommended modifying it as <0.001 or providing a more accurate value with scientific notation.
Author Response
|
1. Introduction - The authors provided thorough background information on the COVID-19 pandemic and the theoretical premise of how proper education and self-protective behaviors could foster disease containment guided by the Protective Motivation Theory (PMT). However, the necessary background information about the development of the intervention per se (i.e., the digital-based self-learned education intervention) is lacking in the Introduction. Please elaborate on the intervention program itself including design, necessary details about the program itself, how it was administered in the institution, etc. |
Modified according to your valuable comment. A paragraph was added to the text and highlighted in green color as follows “Moreover, the recurrent waves of coronavirus infections created a great surge of rumors and misleading information about this novel virus called “infodemic”. Thus, it becomes of utmost importance to develop evidence-based educational interventions (16,17). A theory-based intervention using information and communication technology (ICT) and self-education proves robust evidence in the latest globalization era. Self-learning and digital-based education, as an example of ICT, offer several benefits. They have more flexibility, accessibility, and low cost which can help in cultivating knowledge and behavioral change as guided by self-motivation, not coercive attendance of predesigned sessions (18-20). In this study, the investigators apply this approach guided by the PMT using four stages approach: need assessment, planning, implementation, and evaluation. A peer-reviewed educational content (PowerPoint presentation, and a recorded session using blackboard) was developed after assessing the participants’ educational needs. Then, they were made available to the participants through their official university emails. A synchronized availability of the investigators through Zoom cloud meetings was done weekly, throughout the study duration, to answer their questions and concerns. Besides, a key health message about COVID-19 was shared with them daily to maintain contact, and engagement, boost their self-efficacy and intention for self-protection and turn it into action”. |
3 |
99-117 |
|
2. Abstract - Related to #1. There was a lack of description of the administration of digital self-learned education intervention about COVID-19 in the Methods section of the abstract. |
Modified according to your valuable comment. A paragraph was added in the abstract and highlighted in green color as follows “The educational intervention was designed using four stages approach: need assessment, planning, implementation, and evaluation. A peer-reviewed digital educational content was developed after assessing the participants’ educational needs using a pretest. Then, distributed to them through their university emails. The educational intervention was designed using four stages approach: need assessment, planning, implementation, and evaluation. A peer-reviewed digital educational content was developed after assessing the participants’ educational needs using the pretest. Then, distributed through their university emails. A weekly synchronous Zoom cloud meeting and a daily key health message was shared with them. Finally, the posttest was conducted after two months” |
1 |
15-20 |
|
3. Material and Methods - Consider providing the list of key questions included in the questionnaire as supplemental material (e.g. Part II-IV). It would be also helpful to specify how individual questions were mapped to the pre-specified survey constructs. |
Modified according to your valuable comment. and highlighted in green color in the text as follows: “The instrument ‘s discriminative ability was evaluated using Pearson's correlation coefficient between items and the respective scales. Thus, two items were removed due to low coefficient (<30) and lack of significance (P>0.05). The instrument's construct validity was confirmed by the confirmatory factor analysis which showed a good fit using the comparative fit index (CFI >0.90)” - An English version of the questionnaire was uploaded on the journal system as a supplementary material |
5 |
194-197 |
|
4. Material and Methods - The reviewer was wondering whether the educational intervention (2.5.) should be moved before the survey development (2.3.). |
According to our point of view we firstly develop the survey and assure its validity and reliability to be used for assessing the educational needs of the participants using the pretest. Thus, we can develop a need-based and relevant educational intervention. |
|
|
|
5. Material and Methods - currently the statistical analyses were described under 2.5. Pilot Study and Data Collection. The reviewer would recommend having a standalone section for data analysis to avoid confusion. |
Modified according to your valuable comment and highlighted in green color in the text |
||
|
2.5. Pilot study |
5 |
201-204 |
|
|
2.6. Field work |
5-6 |
205-242 |
|
|
2.7. statistical analysis |
6 |
243-255 |
|
|
6. Were any sociodemographic characteristics analyzed in association with the outcomes of interest related to knowledge and self-protective behaviors?
|
We assured the matched criteria between the study and control groups by chi square test before conducting the intervention to avoid confounding effect. But we did not analyze association of sociodemographic characteristics with the study outcome because our main concern is mainly for evaluating the effectiveness of the educational intervention |
7-8 |
309-311 |
|
Minor:
7. Page 2, para 2: spell out PMT when it first appeared in the main text.
|
Modified according to your valuable comment and highlighted in green color in the text |
2 |
68-69 |
|
8. Please confirm all P values of 0.000. The reviewer recommended modifying it as <0.001 or providing a more accurate value with scientific notation. |
Modified according to your valuable comment and highlighted in green color in the tables. |
|
|

Reviewer 2 Report
This study used a multi-stage sample of participants to assess students' knowledge, self-protective behaviours and the structure of the PMT using an electronic self-administered questionnaire to evaluate the impact of a digital self-learning educational intervention on knowledge and self-protective behaviours among non-healthy students at Saudi SEU. The results showed that the digital PMT-based self-directed learning educational intervention was effective in improving knowledge, protective motivation and self-protective behaviours of non-healthy university students with New Coronary Pneumonia. The study is well founded, well argued, well justified and has some practical value, but there are still the following issues that suggest corrections.
1. In line38-40, "It is formally known as Coronavirus Disease-2019 (COVID-19) and was announced as a global pandemic by the World Health It is formally known as Coronavirus Disease-2019 (COVID-19) and was announced as a global pandemic by the World Health Organization (WHO) in March 2020".
2. The line number after Discussion is not shown.
Author Response
Replay to reviewer 2
Dear reviewer, thanks for your vulnerable comments.
|
Comment |
Replay |
Page |
Line |
|
1. In line38-40, "It is formally known as Coronavirus Disease-2019 (COVID-19) and was announced as a global pandemic by the World Health It is formally known as Coronavirus Disease-2019 (COVID-19) and was announced as a global pandemic by the World Health Organization (WHO) in March 2020".
|
Modified according to your valuable comment in the text and highlighted in green color, as follows “It is formally known as Coronavirus Disease-2019 (COVID-19). It was announced by the World Health Organization (WHO) on 30 January 2020 as a public health emergency and on 11 March 2020 as a global pandemic”
The reference also (number 4) was updated in the reference list and highlighted in green color |
2 |
43-45 |
|
2. The line number after Discussion is not shown. |
I think, this issue is related to the journal system |
|
|
|
|
|
|
|

Round 2
Reviewer 1 Report
Thank you for taking my suggestions into consideration. Please make sure the English version of the questionnaire as supplementary material was referenced in the main text. Otherwise, I do not have additional comments or concerns. It's my great pleasure to review your work!
Author Response
Dear reviewer, thanks for your vulnerable comments.
|
Comment |
Replay |
Page |
Line |
|
Please make sure the English version of the questionnaire as supplementary material was referenced in the main text. |
Done and highlighted in red color. " English version of the used questionnaire is available as supplementary file (please insert hyperlink here on the word file)." " |
5 |
190-191 |
|
Editor comment |
|
|
|
|
(III) Please check that all references are relevant to the contents of the manuscript. |
Done |
14-16 |
|
|
If one of the referees has suggested that your manuscript should undergo |
Although not recommended, Grammarly premium was used to make grammar check and the report is attached. |
|
|
Dear reviewers
We were very fortunate that you reviewed our research. Your comments significantly improved the quality of our research. Thank you very much.